# Real-World Data Validation of NAPOLI-1 Nomogram for the Prediction of Overall Survival in Metastatic Pancreatic Cancer

**DOI:** 10.3390/cancers15041008

**Published:** 2023-02-05

**Authors:** Yung-Yeh Su, Nai-Jung Chiang, Yi-Hsin Yang, Chia-Jui Yen, Li-Yuan Bai, Chang-Fang Chiu, Shih-Chang Chuang, Shih-Hung Yang, Wen-Chi Chou, Jen-Shi Chen, Tai-Jan Chiu, Yen-Yang Chen, De-Chuan Chan, Cheng-Ming Peng, Sz-Chi Chiu, Chung-Pin Li, Yan-Shen Shan, Li-Tzong Chen

**Affiliations:** 1National Institute of Cancer Research, National Health Research Institutes, Tainan 704016, Taiwan; 2Department of Oncology, National Cheng Kung University Hospital, College of Medicine, National Cheng Kung University, Tainan 704302, Taiwan; 3Institute of Clinical Medicine, College of Medicine, National Cheng Kung University, Tainan 704017, Taiwan; 4Center for Cancer Research, Kaohsiung Medical University, Kaohsiung 807377, Taiwan; 5Department of Oncology, Taipei Veterans General Hospital, Taipei 112201, Taiwan; 6School of Medicine, College of Medicine, National Yang Ming Chiao Tung University, Taipei 112304, Taiwan; 7Division of Hematology and Oncology, Department of Internal Medicine, China Medical University Hospital, China Medical University, Taichung 404327, Taiwan; 8School of Medicine, College of Medicine, China Medical University, Taichung 404328, Taiwan; 9Cancer Center, China Medical University Hospital, China Medical University, Taichung 404327, Taiwan; 10Division of General and Digestive Surgery, Department of Surgery, Kaohsiung Medical University Hospital, Kaohsiung 807377, Taiwan; 11Department of Surgery, Faculty of Medicine, Kaohsiung Medical University, Kaohsiung 807378, Taiwan; 12Department of Oncology, National Taiwan University Hospital, Taipei 100229, Taiwan; 13Division of Hematology-Oncology, Department of Internal Medicine, Linkou Chang Gung Memorial Hospital, Taoyuan 333423, Taiwan; 14College of Medicine, Chang Gung University, Taoyuan 33302, Taiwan; 15Division of Hematology-Oncology, Department of Internal Medicine, Kaohsiung Chang Gung Memorial Hospital, Kaohsiung 833401, Taiwan; 16Division of General Surgery, Department of Surgery, Tri-Service General Hospital, National Defense Medical Center, Taipei 114202, Taiwan; 17Department of Surgery, Chung Shan Medical University Hospital, Chung Shan Medical University, Taichung 402306, Taiwan; 18PharmaEngine, Inc., Taipei 104511, Taiwan; 19Division of Clinical Skills Training, Department of Medical Education, Taipei Veterans General Hospital, Taipei 112201, Taiwan; 20Division of Gastroenterology and Hepatology, Department of Medicine, Taipei Veterans General Hospital, Taipei 112201, Taiwan; 21Division of General Surgery, Department of Surgery, National Cheng Kung University Hospital, College of Medicine, National Cheng Kung University, Tainan 704302, Taiwan; 22Department of Internal Medicine, Kaohsiung Medical University Hospital, Kaohsiung Medical University, Kaohsiung 807377, Taiwan

**Keywords:** pancreatic cancer, nal-IRI, nomogram, real-world

## Abstract

**Simple Summary:**

The nomogram derived from the pivotal phase III NAPOLI-1 study could predict the overall survival in gemcitabine-refractory metastatic pancreatic cancer treated with liposomal irinotecan plus fluorouracil and leucovorin. However, the NAPOLI-1 nomogram has not been validated in a real-world setting and therefore the applicability of the NAPOLI-1 nomogram in daily practice remains unknown. In the current study, we validated the NAPOLI-1 nomogram in a multicenter real-world cohort and confirmed that the NAPOLI-1 nomogram could predict the prognosis of gemcitabine-refractory metastatic pancreatic cancer in daily practice and may help clinical decision making. We further found that the relative dose intensity at 6 weeks was an independent prognostic factor beyond the NAPOLI-1 nomogram, which highlighted the importance of optimal dose delivery regardless of the baseline condition.

**Abstract:**

Background: The nomogram derived from the pivotal phase III NAPOLI-1 study demonstrated a significant ability to predict median overall survival (OS) in gemcitabine-refractory metastatic pancreatic ductal adenocarcinoma (PDAC) treated with liposomal irinotecan plus fluorouracil and leucovorin (nal-IRI+5-FU/LV). However, the NAPOLI-1 nomogram has not been validated in a real-world setting and therefore the applicability of the NAPOLI-1 nomogram in daily practice remains unknown. This study aims to evaluate the NAPOLI-1 nomogram in a multicenter real-world cohort. Methods: The NAPOLI-1 nomogram was applied to a previously established cohort of metastatic PDAC patients treated with nal-IRI+5-FU/LV in nine participating centers in Taiwan. Patients were divided into three risk groups according to the NAPOLI-1 nomogram. The survival impact of relative dose intensity at 6 weeks (RDI at 6 weeks) in different risk groups was also investigated. Results: Of the 473 included patients, the median OSs of patients classified as low (*n* = 156), medium (*n* = 186), and high (*n* = 131) risk were 10.9, 6.3, and 4.3 months, respectively (*p* < 0.0001). The survival impact of RDI at 6 weeks remained significant after stratification by risk groups, adjustment with Cox regression, inverse probability weighting, or propensity score matching. Conclusions: Our results support the usefulness of the NAPOLI-1 nomogram for risk stratification in gemcitabine-refractory metastatic PDAC treated with nal-IRI+5-FU/LV in daily practice. We further showed that the RDI at 6 weeks is an independent prognostic factor beyond the NAPOLI-1 nomogram.

## 1. Introduction

Liposomal irinotecan plus 5-fluorouracil/leucovorin (nal-IRI+5-FU/LV) has been shown to improve overall survival (OS) compared to 5-FU/LV alone (median OS 6.1 months vs. 4.2 months, hazard ratio 0.67, *p* = 0.012) in patients with metastatic pancreatic ductal adenocarcinoma (PDAC) who have failed to respond to gemcitabine-based chemotherapy (NAPOLI-1 Study) [1]. Based on the NAPOLI-1 study, nal-IRI+5-FU/LV was approved by the United States Food and Drug Administration (FDA) in October 2015 and to date, nal-IRI+5-FU/LV remains the only approved regimen in patients with gemcitabine-refractory metastatic PDAC. Several studies using real-world data (RWD) assessing the effectiveness of nal-IRI+5-FU/LV in metastatic PDAC have been reported but their clinical outcomes are barely comparable due to the heterogeneity in study populations and different practice patterns. For example, the proportion of Eastern Cooperative Oncology Group Performance Status (ECOG PS) ≥ 2 ranged from 0 to 27.3% in the published real-world data, while 1.9% to 69.6% of patients had a reduced starting dose and 20.5% to 50% of patients required dose modification [2,3,4,5,6,7,8,9,10,11,12,13,14,15]. Such heterogenicity among studies resulted in heterogeneous treatment outcomes, i.e., an objective response rate of 2.9–19.2%, median progression-free survival (PFS) of 2.0–4.5 months, and median overall survival of 4.3–9.4 months.

A nomogram, usually derived from complicated models, is a user-friendly tool to help physicians quickly and easily understand the prognosis of a patient and aid clinical decision-making [16,17,18]. A well-established nomogram will not only help clinicians with risk stratification and decision-making but may also serve as a comparison benchmark to compare different studies. The NAPOLI-1 study-derived nomogram consisted of eight independent dichotomized parameters identified in multivariate Cox regression for overall survival in the NAPOLI-1 population, including baseline Karnofsky performance score, baseline albumin level, baseline neutrophil–lymphocyte ratio, the presence of liver metastasis, baseline CA19-9, stage IV at diagnosis, body mass index (BMI) and receiving nal-IRI+5-FU/LV combination treatment [1,19]. Based on the NALOPI-1 nomogram, a risk score ranging from 0 to 680 was calculated for each patient. The NAPOLI-1 population was then divided into three risk tertiles, low risk, intermediate risk, and high risk, with cut-off point scores of >370, 260–370, and <260, respectively. The NAPOLI-1 nomogram demonstrated significant discriminatory power to predict overall survival. Patients in high-, intermediate-, and low-risk groups had a median OS of 2.9, 5.3, and 8.5 months, respectively. However, the NAPOLI-1 nomogram was not validated in a real-world setting and therefore the applicability of the NAPOLI-1 nomogram in daily practice remains uncertain.

The per-protocol (PP) analysis of the NAPOLI-1 study defined the PP population as relative dose intensity at 6 weeks (RDI at 6 weeks) ≥ 80% and the median OS of the PP population was significantly better than that of non-PP populations (8.9 vs. 4.4 months, respectively) [20]. Our previous real-world multicenter study further set another cut-point of 60% and divided patients into three groups based on their RDI at 6 weeks: >80%, 60–80%, and <60%. We found that RDI at 6 weeks was an independent prognostic factor of OS even after adjustment of covariates by multivariable Cox regression [21]. In this study, we validate the NAPOLI-1 nomogram in the same large real-world multicenter cohort and further address how RDI at 6 weeks impacts the survival of patients in different risk groups.

## 2. Materials and Methods

A real-world multicenter cohort of patients with pancreatic cancer treated with nal-IRI+5-FU/LV has been previously described [21]. The inclusion and exclusion criteria of the current study were set to accord with the NAPOLI-1 population. The inclusion criteria included (1) prior use of gemcitabine; (2) presence of metastatic disease while receiving nal-IRI+5-FU/LV; patients with an initial diagnosis of American Joint Committee on Cancer (AJCC) stage I-III are eligible if they developed metastasis after gemcitabine-based treatment; and (3) Karnofsky performance score ≥ 70 (equivalent to ECOG performance score 0–1) while receiving nal-IRI+5-FU/LV. Patients with the following conditions were excluded: (1) using nal-IRI without 5-FU or in combination with S1; (2) receiving nal-IRI+5-FU/LV plus other agents; and (3) receiving nal-IRI+5-FU/LV before the reimbursement in August 2018. This retrospective study was approved by the Institutional Review Board (IRB) with a waiver of informed consent and followed the Declaration of Helsinki. The IRB approval numbers of each participating institute were as follows: Chang Gung Memorial Hospital, 202100783B0; China Medical University Hospital, CMUH109-REC2-176; Chung Shan Medical University Hospital, CS2-21095; National Cheng Kung University Hospital, A-ER-109-477; National Taiwan University Hospital, 201911042RINC; Kaohsiung Medical University Hospital, KMUHIRB-E(I)-20210150; Taipei Veterans General Hospital, 2021-08-001AC; and Tri-Service General Hospital, B202105057.

Patient baseline characteristics, outcomes, and adverse events were extracted from electronic medical records. Tumor response was evaluated by computed tomography (CT) or magnetic resonance imaging (MRI) every 8–12 weeks at the physician’s discretion. Common Terminology Criteria for Adverse Events (CTCAE) version 4.0.3 was used to evaluate the adverse events. The risk score of each patient was calculated from a previously published NAPOLI-1 nomogram consisting of eight dichotomized parameters: baseline Karnofsky performance score ≥ 90 vs. <90 (equivalent to ECOG PS 0 vs. ≥1); baseline albumin ≥ 4 g/dL vs. <4 g/dL; neutrophil–lymphocyte ratio ≤ 5 vs. >5; absence vs. presence of liver metastasis; baseline CA19-9 ≤ 1542 IU/mL vs. >1542 IU/mL; stage I to III vs. stage IV at diagnosis; body mass index (BMI) > 25 kg/m^2^ vs. ≤25 kg/m^2^; nal-IRI+5-FU/LV vs. nal-IRI or 5-FU/LV monotherapy (all patients in our study received nal-IRI+5-FU/LV) [19]. Patients were then classified as low (score points > 370), intermediate (score points 260–370) and high risk (score points < 260) according to the original NAPOLI-1 nomogram report.

Descriptive statistics are presented as medians or percentages. The difference in proportions between groups was compared by Fisher’s exact test. The Kolmogorov–Smirnov test was used to check the normality of data distribution.

The reverse Kaplan–Meier method was used to estimate the median duration of follow-up. The definition of PFS was the time between the beginning of nal-IRI+5-FU/LV to the date of either radiological or clinical progression, death, intolerance, loss to follow-up, or data cut-off. OS was defined as the interval between the beginning of nal-IRI+5-FU/LV and death, loss to follow-up, or data cut-off. Intolerance, loss to follow-up, or data cut-off will be censored in PFS data while the loss to follow-up or data cut-off will be censored in OS data. The Kaplan–Meier method was applied for the calculation of PFS and OS while the log-rank test was used for survival comparison between groups.

Three parameters in the NAPOLI-1 nomogram, albumin, neutrophil–lymphocyte ratio, and baseline CA19-9, were not available in some patients and were imputed by using the R multivariate imputation by chained equation (MICE) package [22]. The area under the time-dependent receiver operating characteristic curve (AUC) with inverse probability of censoring weighting (IPCW) technique was estimated to evaluate the prediction accuracy of the NAPOLI-1 nomogram. In terms of investigating the effect of cumulative dose, Cox regression was used to estimate the hazard ratio (HR) and 95% CI with the nomogram score as an adjusted covariate. Propensity score weighting and matching methods were also applied to further balance the distribution of nomogram scores among cumulative dose groups. The inverse probability of weights (IPW) was computed from the generalized propensity scores (GPSs) by using multinomial logistic regression with cumulative dose groups as the outcome variable and the nomogram as the covariate. The 3 groups were matched at 1:1:1 with the same nomogram score. When more than one participant had the same score, patients were randomly selected. We also conducted these statistical analyses on patients without any missing values (*n* = 264) as a sensitivity analysis. All variables with *p* < 0.05 were statistically significant. All statistical analyses were performed using R version 4.0.5 (R Core Team, Vienna, Austria) and SAS version 9.4 (SAS Institute Inc., Cary, NC, USA).

## 3. Results

### 3.1. Patient Demographics

A total of 696 patients treated with nal-IRI+5-FU/LV were identified from 9 participating centers in Taiwan, as previously described [21]. Among them, 523 patients met the inclusion criteria. After excluding the 50 patients whose treatment had a combination that included oxaliplatin (*n* = 19), immunotherapy (*n* = 4) or other agents (*n* = 3), or whose treatment used S1 instead of 5-FU (*n* = 3) and those who received nal-IRI before reimbursement in Aug. 2018 (*n* = 21), finally, 473 patients fulfilled the inclusion and exclusion criteria and were included in the current study (Figure 1).

Using the NAPOLI-1 nomogram for risk classification, 156, 186, and 131 patients were classified as belonging to low, intermediate, and high-risk groups, respectively. As classification parameters, four baseline characteristics in the nomogram including disease stage at diagnosis, baseline albumin, CA 19.9, and liver metastasis were different among the three risk groups (Table 1). Otherwise, all remaining baseline characteristics were comparable between low, intermediate, and high-risk groups.

As of the data cutoff on 31 December 2020, the median duration of follow-up was 13.1 months (interquartile range, IQR 7.0–20.6 months). The median PFS and OS in low-, intermediate-, and high-risk groups were significantly different. The median PFSs in the corresponding risk groups were 4.8 (95% CI, 3.5–6.8), 3.2 (95% CI, 2.7–4.7), and 2.3 (95% CI, 2.0–2.8) months, respectively (log-rank *p* < 0.0001, Figure 2A); while the median OSs were 10.9 (95% CI, 9.3–12.5), 6.3 (95% CI, 5.7–7.3) and 4.3 (95% CI, 3.8–5.5) months, respectively (log-rank *p* < 0.0001, Figure 2B).

In the sensitivity analysis which excluded 209 patients with missing values in the nomogram, the distribution of NAPOLI-1 nomogram risk classification, median PFS, and median OS of the remaining 264 patients without missing value imputation were similar to those of the entire cohort (Appendix A).

### 3.2. Model Performance

In the current study, the AUCs of the NAPOLI-1 nomogram for median OS prediction range from 0.698 to 0.738 (Figure 3A,C). Most of the parameters used in the NAPOLI-1 nomogram remain significant in the multivariable analysis for the current RWD population (Appendix A). Another multivariable analysis was performed to further explore whether any factor other than the NAPOLI-1 nomogram had a significant impact on survival by selecting a covariate with a *p*-value < 0.05 in univariate analysis. RDI at 6 weeks was found to be the most significant factor beyond the NAPOLI-1 nomogram (Appendix A). Adding RDI at 6 weeks to the NAPOLI-1 nomogram improved the AUC mainly at 3 months and 6 months, which reached values of 0.822 and 0.731, respectively (Figure 3B,D). In the sensitivity analysis, the AUC of 264 patients without missing value imputation was similar to that of the entire cohort (Appendix A).

### 3.3. Relative Dose Intensity at 6 Weeks Is an Independent Prognostic Factor

The distribution of cumulative dose during the first 6 weeks was similar among different risk groups (Figure 4A). The distribution of nomogram risk scores was similar among the three dose intensity groups (Figure 4B). Linear regression demonstrated no statistically significant correlation between nomogram risk score and 6-week cumulative dose (Spearman’s correlation coefficient = 0.041, *p* = 0.37) (Figure 4C). All results were consistent in that RDI at 6 weeks was independent of the baseline condition.

The survival of patients with different RDI at 6 weeks was significantly different across three NAPOLI-1 nomogram risk groups (Figure 5A–C). The survival difference remained significant after adjustment with nomogram risk score, age, and gender by Cox regression analysis (<60% vs. >80%, HR 1.975, 95% CI 1.461–2.670) (Table 2). The survival difference among patients with different RDI at 6 weeks remained significant in the inverse probability of weighting survival analysis (<60% vs. >80%, HR 1.800, 95% CI 1.547–2.095, Figure 5D) or after 1:1:1 three-group propensity score matching by nomogram risk score (<60% vs. >80%, HR 3.091, 95% CI 1.643–5.815, Figure 5E).

In the sensitivity analysis, the dose distribution, survival difference before and after covariate adjustment, stratification, weighting, and matching in the 264 patients without missing value imputation were quite similar to those of the entire cohort (Appendix A).

### 3.4. Real-World Safety Profile

The most common ≥ grade 3 hematological toxicities were neutropenia (23.3%) and anemia (19.5%); while the most common non-hematological toxicities were hypokalemia (12.7%) and increased bilirubin (7.4%) (Table 3). There was no significant difference in safety profiles among the three different risk groups.

## 4. Discussion

Our study is the first study to evaluate and validate the NAPOLI-1 nomogram in a real-world setting and support its use as a risk stratification tool, which will not only help physicians and patients to make decisions but can also be implemented in the comparison of results among parallel studies, including real-world data and clinical trials.

In the per-protocol (PP) analysis of the NAPOLI-1 study, patients in PP populations (defined as RDI at 6 weeks ≥ 80%) had significantly better median OS as compared to non-PP populations (8.9 vs. 4.4 months, respectively) [20]. One may argue that the longer survival of patients with higher RDI at 6 weeks might result from better baseline condition as reflected by the higher percentage of patients with Karnofsky performance score ≥ 90 in PP than in non-PP populations (62.1% and 49%, respectively). In our previous study, we showed that RDI at 6 weeks remained an important prognostic factor after confounder adjustment [21]. In the current study, there was no significant correlation between NAPOLI-1 nomogram risk score, an indicator of baseline condition, and RDI at 6 weeks (Figure 3C). In addition, RDI at 6 weeks significantly impacted the survival of different risk groups (Figure 5A–C), even after weighting (Figure 4D), matching (Figure 4E), and Cox regression adjustment (Table 2). Through different methodologies, the results highlighted the importance of dose delivery regardless of the baseline condition.

In the post hoc analysis of the NAPOLI-1 study, the median OS of Asian patients receiving nal-IRI+5-FU/LV was 8.9 months compared to the 6.1 months of the entire nal-IRI+5-FU/LV-treated population [23]. Of note, the median OS of patients with a low and intermediate risk score in the current study was numerically better than that of patients with nal-IRI+5-FU/LV in the NAPOLI-1 study (10.9 vs. 9.0 months and 6.3 vs. 5.4 months, respectively), but similar survival for patients with a high risk score (4.3 vs. 4.3 months) [19]. In a recent population pharmacokinetic study, Asian patients had a significantly higher average concentration of un-encapsulated SN38 (uSN38 C_avg_), a pharmacokinetic parameter associated with prolonged OS, compared to Caucasians in the nal-IRI+5-FU/LV arm of the NAPOLI-1 study [24]. Interestingly, the uSN38 concentration was similar in Caucasian patients with and without *UGT1A1*28/*28* homozygosity, but significantly higher in Asian patients with *UGT1A1*6/*6* homozygosity or *UGT1A1*6/*28* compound heterozygosity than in those with *UGT1A1*6* single heterozygosity or wild-type individuals [24,25]. Since the allele frequency of *UGT1A1*6* only ranged from 15–30% with *UGT1A1*6* homozygosity or compound heterozygosity in 9–10% of the Asian population, the mechanisms for the ethnic differences in nal-IRI pharmacokinetics and survival after nal-IRI+5-FU/LV are likely multifactorial and warrant further investigation.

This study has some limitations. First, three parameters in the nomogram, baseline albumin, neutrophil–lymphocyte ratio, and baseline CA 19-9 level were not available in all patients and imputed by the MICE procedure. The handling of missing values is a huge discipline and each method had its advantages and disadvantages. Originally proposed by Donald Rubin in 1987, the multiple imputation procedure has been greatly improved and become more widely used even in clinical trials published in *The Lancet* and *The New England Journal of Medicine* [26,27,28]. However, the results from the sensitivity analyses of the 264 patients without missing values were consistent with all study results, which justifies the imputation procedure. Second, in the original NAPOLI-1 nomogram, nal-IRI+5-FU/LV treatment was one of the eight dichotomized parameters. The inclusion of such a parameter can be redundant if the goal of the study was intra-study risk categorization for a patient cohort with homogenous treatment. On the other hand, it is reasonable to do so if inter-studies survival outcome comparison is the main theme, such as RWD versus clinical trial results in this study. Third, although all baseline characteristics other than ethnicity were comparable between the current study and NAPOLI-1, the practice pattern was very different in that more patients in our cohort had prior exposure to fluorouracil-containing treatment than did participants in NAPOLI-1 (77.4% vs. 42.7%, respectively). Due to the delayed reimbursement of nab-paclitaxel and FOLFIRINOX, the gemcitabine/fluorouracil-based triplet, including the GOFL regimen (gemcitabine, oxaliplatin, 5-FU, and leucovorin) and SLOG regimen (S1, leucovorin, oxaliplatin, and gemcitabine), were among the most commonly used first-line regimens for advanced pancreatic cancer in Taiwan before 2020 [29,30,31,32,33,34]. Although the current study suggests previous more aggressive gemcitabine-based treatments, such as GOFL and SLOG, might not impact the therapeutic efficacy of second-line nal-IRI+5-FU/LV, the issue still deserves further exploration with NAPOLI-1 nomogram risk group-stratified comparison.

## 5. Conclusions

In the current study, we employed several strategies including stratification, multivariable adjustment, weighting, and matching methods. The results were consistent among these methods, which indicates the robustness of our findings in using the NAPOLI-1 nomogram as a measurable confounding factor. Our study not only validated the NAPOLI-1 nomogram for risk stratification in a real-world setting but also identified RDI at 6 weeks as an independent prognostic factor beyond the NAPOLI-1 nomogram.

## Figures and Tables

**Figure 1 cancers-15-01008-f001:**
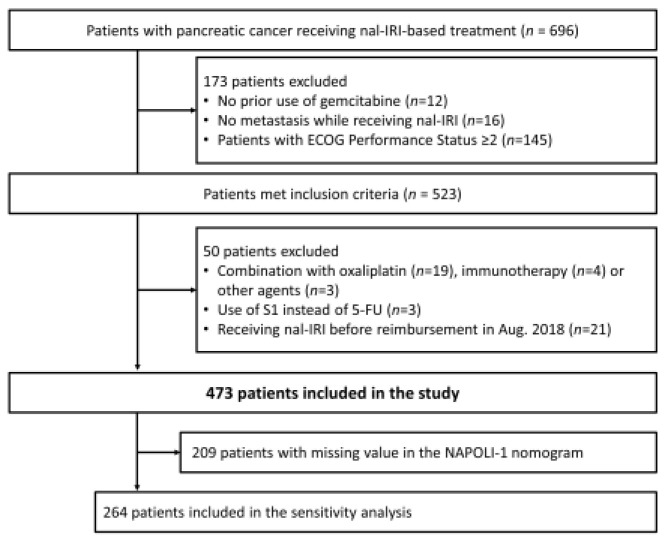
Study schema.

**Figure 2 cancers-15-01008-f002:**
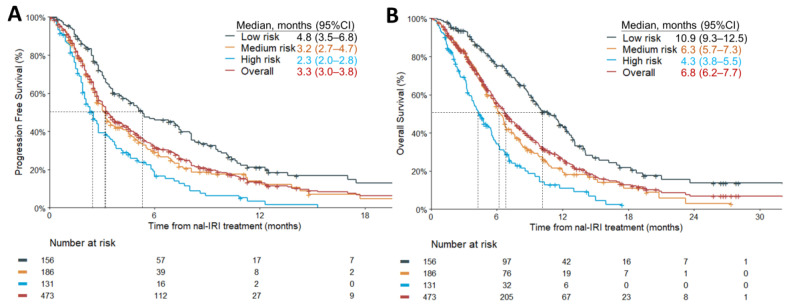
Survival analysis. Kaplan–Meier plot of progression-free survival (**A**) and overall survival (**B**) of the entire cohort and different risk groups.

**Figure 3 cancers-15-01008-f003:**
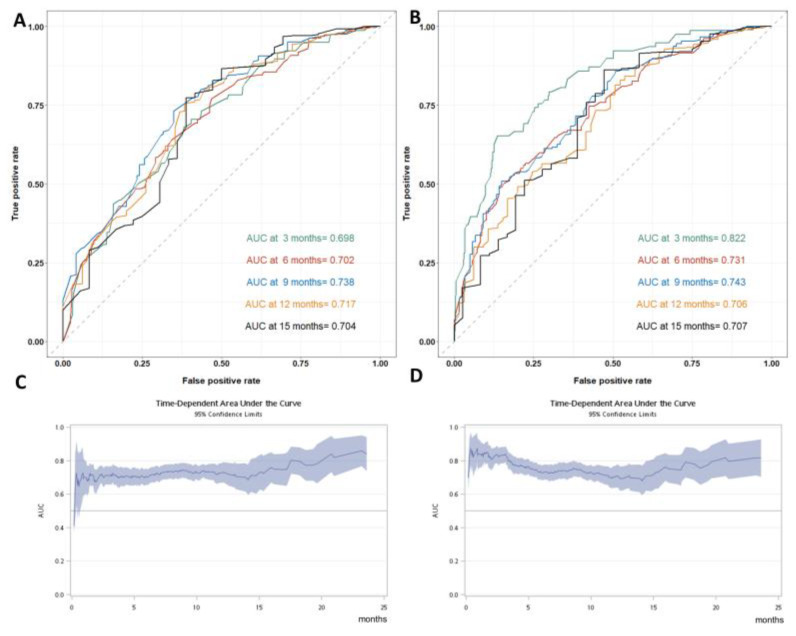
Model Performance. The area under the receiver operating characteristic curve (AUC) at 3, 6, 9, 12 and 15 months of the NAPOLI-1 nomogram (**A**) and of the NAPOLI-1 nomogram with relative dose intensity at 6 weeks (**B**). Time-dependent AUC of the NAPOLI-1 nomogram (**C**) and of the NAPOLI-1 nomogram with relative dose intensity at 6 weeks (**D**).

**Figure 4 cancers-15-01008-f004:**
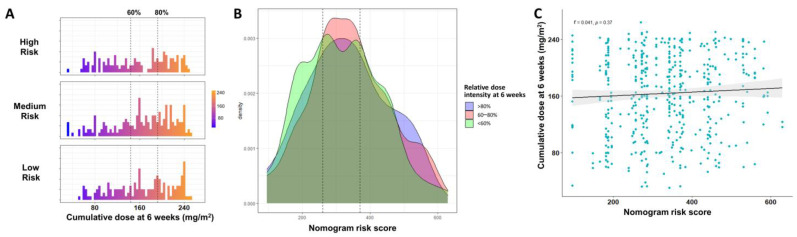
Dose distribution. (**A**) Distribution of cumulative dose at 6 weeks in different NAPOLI-1 nomogram risk groups. (**B**) Distribution of NAPOLI-1 nomogram risk score in different groups of relative dose intensity at 6 weeks. (**C**) Spearman correlation of nomogram risk score and cumulative dose at 6 weeks. Each dot represents one patient.

**Figure 5 cancers-15-01008-f005:**
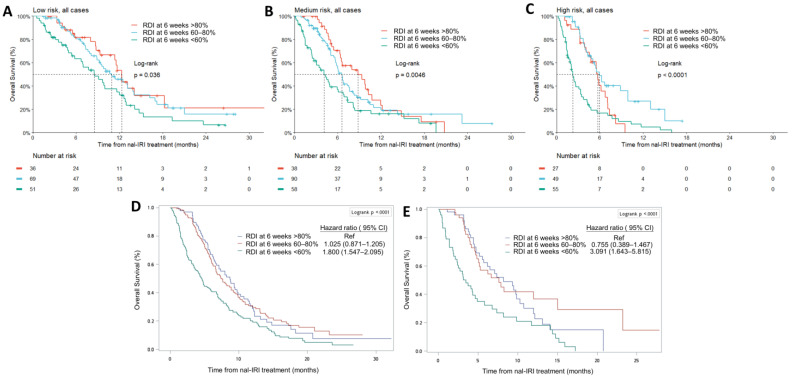
Overall survival. Kaplan–Meier plot in low-risk group (**A**), medium-risk group (**B**), and high-risk group (**C**). (**D**) Overall survival weighted by inverse probability of weights (IPW) in different cumulative dose groups. (**E**) Overall survival after 1:1:1 three-group matching by NAPOLI-1 nomogram risk score in different dose intensity groups (*n* = 53 in each group).

**Table 1 cancers-15-01008-t001:** Baseline characteristics in different risk groups.

Risk Group	Low	Intermediate	High	Overall
	(*n* = 156)	(*n* = 186)	(*n* = 131)	(*n* = 473)
Gender				
Female	67 (42.9%)	83 (44.6%)	48 (36.6%)	198 (41.9%)
Male	89 (57.1%)	103 (55.4%)	83 (63.4%)	275 (58.1%)
Age, median (range)	62.5 (27–82)	63.0 (34–86)	63 (33–86)	63 (27–86)
Disease stage at diagnosis				
Stage I-III	108 (69.2%)	40 (21.5%)	6 (4.6%)	154 (32.6%)
Stage IV	48 (30.8%)	146 (78.5%)	125 (95.4%)	319 (67.4%)
Primary tumor location				
Head	90 (57.7%)	95 (51.1%)	64 (48.9%)	249 (52.6%)
Body	34 (21.8%)	52 (28.0%)	22 (16.8%)	108 (22.8%)
Tail	24 (15.4%)	32 (17.2%)	37 (28.2%)	93 (19.7%)
Body + Tail	7 (4.5%)	6 (3.2%)	7 (5.3%)	20 (4.2%)
Head + Body or Tail	1 (0.6%)	1 (0.5%)	1 (0.8%)	3 (0.6%)
Albumin				
<4	45 (28.8%)	69 (37.1%)	65 (49.6%)	179 (37.8%)
≥4	55 (35.3%)	37 (19.9%)	22 (16.8%)	114 (24.1%)
Not checked	56 (35.9%)	80 (43.0%)	44 (33.6%)	180 (38.1%)
Number of metastatic sites				
1	117 (75.0%)	90 (48.4%)	47 (35.9%)	254 (53.7%)
2	34 (21.8%)	66 (35.5%)	41 (31.3%)	141 (29.8%)
3	5 (3.2%)	27 (14.5%)	33 (25.2%)	65 (13.7%)
≥4	0 (0%)	3 (1.6%)	10 (7.6%)	13 (2.7%)
Site of metastasis				
Liver	69 (44.2%)	125 (67.2%)	125 (95.4%)	319 (67.4%)
Lung	35 (22.4%)	47 (25.3%)	40 (30.5%)	122 (25.8%)
Peritoneum	38 (24.4%)	55 (29.6%)	37 (28.2%)	130 (27.5%)
CA-19.9				
<40 U/mL	37 (23.7%)	26 (14.0%)	6 (4.6%)	69 (14.6%)
≥40 U/mL	109 (69.9%)	138 (74.2%)	99 (75.6%)	346 (73.2%)
Not checked	10 (6.4%)	22 (11.8%)	26 (19.8%)	58 (12.3%)
Prior treatment				
Gemcitabine-containing	156 (100%)	186 (100%)	131 (100%)	473 (100%)
Fluorouracil-containing	112 (71.8%)	150 (80.6%)	104 (79.4%)	366 (77.4%)
Irinotecan-containing	16 (10.3%)	28 (15.1%)	20 (15.3%)	64 (13.5%)
Platinum-containing	59 (37.8%)	86 (46.2%)	73 (55.7%)	218 (46.1%)
Taxane-containing	47 (30.1%)	56 (30.1%)	41 (31.3%)	144 (30.4%)
Prior lines of systemic treatment †				
0	2 (1.3%)	5 (2.7%)	0 (0%)	7 (1.5%)
1	101 (64.7%)	110 (59.1%)	84 (64.1%)	295 (62.4%)
≥2	53 (34.0%)	71 (38.2%)	47 (35.9%)	171 (36.2%)
Operation history				
No surgery	74 (47.4%)	119 (64.0%)	88 (67.2%)	281 (59.4%)
Whipple operation	26 (16.7%)	34 (18.3%)	17 (13.0%)	77 (16.3%)
Distal pancreatectomy	20 (12.8%)	13 (7.0%)	9 (6.9%)	42 (8.9%)
Total pancreatectomy	3 (1.9%)	4 (2.2%)	3 (2.3%)	10 (2.1%)
Other procedures	33 (21.2%)	16 (8.6%)	14 (10.7%)	63 (13.3%)
Interval between the last therapy and nal-IRI+5-FU/LV				
Median (IQR)	0.754 (0.475–1.28)	0.672 (0.459–1.08)	0.689 (0.459–1.11)	0.689 (0.459–1.15)
Not recorded	21 (13.5%)	28 (15.1%)	16 (12.2%)	65 (13.7%)

IQR: interquartile range; † 0 indicated only adjuvant gemcitabine without other systemic treatment before nal-IRI+5-FU/LV.

**Table 2 cancers-15-01008-t002:** NAPOLI-1 nomogram and multivariable Cox regression in the overall population (*n* = 473).

Model 1 NAPOLI-1 Nomogram			Model 2 NAPOLI-1 Nomogram and Cumulative Dose		
Parameter	HR (95%CI)	*p*-Value	Parameter	HR (95%CI)	*p*-Value
Nomogram risk score	0.996 (0.995–0.997)	<0.0001	Nomogram risk score	0.996 (0.995–0.997)	<0.0001
Age	1.009 (0.996–1.023)	0.1607	RDI_6-week_ > 80%	Reference	-
Gender: male	1.406 (1.118–1.766)	0.0034	60–80%	1.050 (0.777–1.419)	0.9148
			<60%	1.975 (1.461–2.670)	<0.0001
			Age	1.008 (0.994–1.021)	0.2786
			Gender: male	1.287 (1.022–1.620)	0.0317

RDI: relative dose intensity.

**Table 3 cancers-15-01008-t003:** Adverse effects in different NAPOLI-1 nomogram risk groups.

Risk Group	Low	Intermediate (*n* = 186)	High	Overall
	(*n* = 156)		(*n* = 131)	(*n* = 473)
Neutropenia				
All-grade	69 (44.2%)	76 (40.9%)	52 (39.7%)	197 (41.6%)
≥grade 3	36 (23.1%)	44 (23.7%)	30 (22.9%)	110 (23.3%)
Febrile neutropenia	4 (2.6%)	7 (3.8%)	5 (3.8%)	16 (3.4%)
Not recorded	4 (2.1%)	2 (1.2%)	3 (2.4%)	9 (1.9%)
Anemia				
All-grade	83 (53.2%)	121 (65.1%)	97 (74.0%)	301(63.6%)
≥grade 3	23 (14.7%)	37 (19.9%)	32 (24.4%)	92 (19.5%)
Not recorded	2 (1.1%)	2 (1.2%)	2 (1.6%)	6 (1.3%)
Thrombocytopenia				
All-grade	26 (16.7%)	47 (25.3%)	42 (32.1%)	115 (24.3%)
≥grade 3	5 (3.2%)	10 (5.4%)	9 (6.9%)	24 (5.1%)
Not recorded	3 (1.6%)	1 (0.6%)	2 (1.6%)	6 (1.3%)
AST or ALT increased				
All-grade	43 (27.6%)	61 (32.8%)	40 (30.5%)	144 (30.4%)
≥grade 3	6 (3.8%)	7 (3.8%)	0 (0%)	13 (2.7%)
Not recorded	40 (21.4%)	44 (27.2%)	33 (26.6%)	117 (24.7%)
Blood bilirubin increased				
All-grade	20 (12.8%)	47 (25.3%)	35 (26.7%)	102 (21.6%)
≥grade 3	10 (6.4%)	15 (8.1%)	10 (7.6%)	35 (7.4%)
Not recorded	23 (12.3%)	8 (4.9%)	9 (7.3%)	40 (8.5%)
Creatinine increased				
All-grade	28 (17.9%)	30 (16.1%)	22 (16.8%)	80 (16.9%)
≥grade 3	0 (0%)	2 (1.1%)	1 (0.8%)	3 (0.6%)
Not recorded	16 (8.6%)	3 (1.9%)	6 (4.8%)	25 (5.3%)
Hypokalemia				
All-grade	44 (28.2%)	61 (32.8%)	47 (35.9%)	152 (32.1%)
≥grade 3	21 (13.5%)	21 (11.3%)	18 (13.7%)	60 (12.7%)
Not recorded	42 (22.5%)	34 (21.0%)	29 (23.4%)	105 (22.2%)
Fatigue				
All-grade	64 (41.0%)	84 (45.2%)	63 (48.1%)	211 (44.6%)
≥ grade 3	1 (0.6%)	5 (2.7%)	2 (1.5%)	8 (1.7%)
Not recorded	12 (6.4%)	13 (8.0%)	10 (8.1%)	35 (7.4%)
Vomiting				
All grade	62 (39.7%)	78 (41.9%)	46 (35.1%)	186 (39.3%)
≥grade 3	5 (3.2%)	8 (4.3%)	2 (1.5%)	15 (3.2%)
Not recorded	3 (1.6%)	2 (1.2%)	4 (3.2%)	9 (1.9%)
Diarrhea				
All-grade	53 (34.0%)	55 (29.6%)	34 (26.0%)	142 (30.0%)
≥grade 3	3 (1.9%)	5 (2.7%)	5 (3.8%)	13 (2.7%)
Not recorded	5 (2.7%)	3 (1.9%)	5 (4.0%)	13 (2.7%)
Hypoalbuminemia				
All-grade	29 (18.6%)	50 (26.9%)	44 (33.6%)	123 (26.0%)
≥grade 3	1 (0.6%)	5 (2.7%)	0 (0%)	6 (1.3%)
Not recorded	28 (15.0%)	22 (13.6%)	15 (12.1%)	65 (13.7%)

## Data Availability

The data presented in this study are available on request from the corresponding author.

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
