# Peer review of "Real-World Data Validation of NAPOLI-1 Nomogram for the Prediction of Overall Survival in Metastatic Pancreatic Cancer"

_cancers, 2023, doi:10.3390/cancers15041008_

Round 1

Reviewer 1 Report

Topic addressed with great originality and above all relevance of the problem of a severely aggressive cancer.

Excellent setting of materials and methods: concise and complete.

Execellent results of tables. Great scientific paper.

Author Response

Thank you for your review and encouragement. We will keep working on this severely aggressive cancer.

Reviewer 2 Report

Thank you for validating the NAPOLI-1 nomogram with real-world data.

  The issue is whether this nomogram needed to be validated again.  The article accomplishes what it set it to do.

Some relevant citations:

Chen LT, Macarulla T, Blanc JF, et al. Nomogram for Predicting Survival in Patients Treated with Liposomal Irinotecan Plus Fluorouracil and Leucovorin in Metastatic Pancreatic Cancer. Cancers (Basel). 2019;11(8):1068. Published 2019 Jul 28. doi:10.3390/cancers11081068

Hsu CC, Liu KH, Chang PH, et al. Development and validation of a prognostic nomogram to predict survival in patients with advanced pancreatic cancer receiving second-line palliative chemotherapy. J Gastroenterol Hepatol. 2020;35(10):1694-1703. doi:10.1111/jgh.14926

Author Response

Thank you for your review and encouragement. The above relevant articles were cited accordingly.

Reviewer 3 Report

I am very sorry but this article may contain interesting data but then it really needs to be written and presented in a scientifically acceptable way.

Some notes and ambiguities below

Please explain the NAPOLI-I study, it is not usual to expect all the readers to know all the clinical studies in pancreatic cancer.

Liposomal irinotecan plus 5-fluorouracil/leucovorin (nal-IRI+5-FU/LV) was the only approved agent in patients with metastatic pancreatic ductal adenocarcinoma 74 (mPDAC) who have failed to gemcitabine-based chemotherapy.

I would suggest Liposomal irinotecan plus 5-fluorouracil/leucovorin (nal-IRI+5-FU/LV) has been shown to improve overall survival in patients with metastatic pancreatic ductal adenocarcinoma 74 (mPDAC) who have failed to gemcitabine-based chemotherapy (NAPOLI-! Study). 

Page 2 line 89- 95. Please first describe what the nomograms consist of followed by the demonstration to predict power.

NAPOLI-1 population was then divided into 96 three-risk tertiles, low risk, intermediate risk, and high risk, with a cut-off point scores of > 370, 260-370, and <260, respectively. Please provide inside how did the score come about.

Page 3 line 105

It is unclear to me whether this is another real-world multicenter cohort.Is this a different cohort than the cohort in which the nomogram was made?

Page 3 line 102-6

In the previous study the authors concluded: Our results support the use of nal-IRI+5-FU/LV in gemcitabine-refractory mPDAC and suggest that a lower starting dose followed by a re-escalation strategy could achieve clinical outcomes comparable to those with standard starting doses in real-world practice. However here they conclude: “Our previous study identified the median OS was significantly different among patients with relative dose intensity at 6-week (RDI at 6-week) >80%, 60-80% and <60% even after adjustment of covariates by multivariable Cox regression”. This is unclear to me as is what they exactly mean with an RDI.

Page 3 line 110

The inclusion and exclusion criteria 110 of the current study were set to accord with the NAPOLI-1 nomogram. It is unusual to accord in- and exclusion criteria with a nomogram. 

Page 3 line 126 

Tumor response was evaluated by computed tomography 126 (CT) or Magnetic Resonance Imaging (MRI) every 8-12 weeks OR at physician discretion.

Table 1

Which staging system was used? In the regular staging system: stage I is (borderline)resectable disease. So why were those patients included?

Please also use a formula for albumin. 

Where the distant lymph node histopathological confirmed. If not leave them out of the table.

Prior treatment group and operation history? This makes it very confusing for me. Since most patients had stage I-III disease it is surprising to see why just a few patients were operated on. Furthermore, the most performed surgical procedures were “Other procedures”.

Author Response

Reviewer 3

Comments and Suggestions for Authors

I am very sorry but this article may contain interesting data but then it really needs to be written and presented in a scientifically acceptable way.

Some notes and ambiguities below

Reviewer 3: Please explain the NAPOLI-I study, it is not usual to expect all the readers to know all the clinical studies in pancreatic cancer.

I would suggest Liposomal irinotecan plus 5-fluorouracil/leucovorin (nal-IRI+5-FU/LV) has been shown to improve overall survival in patients with metastatic pancreatic ductal adenocarcinoma (mPDAC) who have failed to gemcitabine-based chemotherapy (NAPOLI-! Study).

Author reply:

Thanks for the suggestion. We revised the sentence accordingly with a summary of the NAPOLI-1 study in the introduction section.

Manuscript revision (page 2 lines 73-80):

Liposomal irinotecan plus 5-fluorouracil/leucovorin (nal-IRI+5-FU/LV) has been shown to improve overall survival (OS) compared to 5-FU/LV alone (median OS 6.1 months versus 4.2 months, hazard ratio 0.67, p=0.012) in patients with metastatic pancreatic ductal adenocarcinoma (PDAC) who have failed to gemcitabine-based chemotherapy (NAPOLI-1 Study).[1] Based on the NAPOLI-1 study, nal-IRI+5-FU/LV was approved by the United States Food and Drug Administration (FDA) in October 2015 and to date, nal-IRI+5-FU/LV remains the only approved regimen in patients with gemcitabine-refractory metastatic PDAC.

Reviewer 3:

Page 2 line 89- 95. Please first describe what the nomograms consist of followed by the demonstration to predict power.

NAPOLI-1 population was then divided into three-risk tertiles, low risk, intermediate risk, and high risk, with a cut-off point scores of > 370, 260-370, and <260, respectively. Please provide inside how did the score come about.

Author reply:

Thanks for the constructive suggestion. We rearranged the sentence and added the information about nomogram score calculation.

Manuscript revision (page 2-3 lines 94-106):

The NAPOLI-1 study-derived nomogram consisted of eight independent dichotomized parameters identified in multivariate Cox’s regression for overall survival in the NAPOLI-1 population, including baseline Karnofsky performance score, baseline albumin level, baseline neutrophil-lymphocyte ratio, the presence of liver metastasis, baseline CA19-9, stage IV at diagnosis, body mass index (BMI) and receiving nal-IRI+5-FU/LV combination treatment.[1, 19] Based on the NALOPI-1 nomogram, a risk score ranging from 0 to 680 was calculated for each patient. The NAPOLI-1 population was then divided into three risk tertiles, low risk, intermediate risk, and high risk, with a cut-off point score of > 370, 260-370, and <260, respectively. The NAPOLI-1 nomogram demonstrated significant discrimination power to predict overall survival. Patients in high, intermediate, and low-risk groups had a corresponding median OS of 2.9, 5.3, and 8.5 months, respectively.

Reviewer 3:

Page 3 line 105

It is unclear to me whether this is another real-world multicenter cohort. Is this a different cohort than the cohort in which the nomogram was made?

Author reply:

Yes, this is a different cohort. The NAPOLI-1 nomogram was made from the NAPOLI-1 study. Our cohort is a real-world multicenter cohort in Taiwan. Therefore, our study could serve as a validation cohort for the NAPOLI-1 nomogram.

Reviewer 3:

Page 3 line 102-6

In the previous study the authors concluded: Our results support the use of nal-IRI+5-FU/LV in gemcitabine-refractory mPDAC and suggest that a lower starting dose followed by a re-escalation strategy could achieve clinical outcomes comparable to those with standard starting doses in real-world practice. However here they conclude: “Our previous study identified the median OS was significantly different among patients with relative dose intensity at 6-week (RDI at 6-week) >80%, 60-80% and <60% even after adjustment of covariates by multivariable Cox regression”. This is unclear to me as is what they exactly mean with an RDI.

Author reply:

The approved dose of nal-IRI was 80 mg/m2 (equivalent to 70 mg/m2 of irinotecan base) every 2 weeks. Therefore, the cumulative at 6 weeks will be 240 mg/m2. The per-protocol (PP) analysis of the NAPOLI-1 study defined the PP population as relative dose intensity at 6 weeks (RDI at 6 weeks) ≥80%, namely ≥192 mg/m2. The median OS of the PP population was significantly better than non-PP populations, 8.9 versus 4.4 months, respectively. Based on the PP analysis of the NAPOLI-1 study, our previous real-world multicenter study further set another cut-point of 60% and divided patients into 3 groups, RDI at 6 weeks>80%, 60-80%, and <60%. We found RDI at 6 weeks was an independent prognostic factor of OS even after adjustment of co-variates by multivariable Cox regression. In this study, we further address this issue in another way, using the NAPOLI-1 nomogram as a co-variant adjustment. We made some revisions in the introduction section to make this context clearer.

Manuscript revision (page 3 lines 108-116):

The per-protocol (PP) analysis of the NAPOLI-1 study defined the PP population as relative dose intensity at 6 weeks (RDI at 6 weeks) ≥80% and the median OS of PP population was significantly better than non-PP populations, 8.9 versus 4.4 months, respectively.[20] Our previous real-world multicenter study further set another cut-point of 60% and divided patients into 3 groups, RDI at 6 weeks>80%, 60-80%, and <60%. We found RDI at 6 weeks was an independent prognostic factor of OS even after adjustment of covariates by multivariable Cox regression.[21] In this study, we vali-date the NAPOLI-1 nomogram in the same large real-world multicenter cohort and further address how RDI at 6 weeks impacts the survival of patients in different risk groups.

Reviewer 3:

Page 3 line 110

The inclusion and exclusion criteria of the current study were set to accord with the NAPOLI-1 nomogram. It is unusual to accord in- and exclusion criteria with a nomogram.

Author reply:

Thanks for the kind proofreading. We used “NAPOLI-1 population” instead.

Manuscript revision (page 3 line 120):

The inclusion and exclusion criteria of the current study were set to accord with the NAPOLI-1 population.

Reviewer 3:

Page 3 line 126

Tumor response was evaluated by computed tomography (CT) or Magnetic Resonance Imaging (MRI) every 8-12 weeks OR at physician discretion.

Table 1

Which staging system was used? In the regular staging system: stage I is (borderline)resectable disease. So why were those patients included?

Author reply:

We used the commonly used AJCC staging system. In Table 1, patients with initial stage I-III were included because they developed metastases after gemcitabine-based treatment. We made some amendments to the inclusion criteria.

Manuscript revision (page 3 lines 120-124):

The inclusion criteria included 1) prior use of gemcitabine; 2) presence of metastatic disease while receiving nal-IRI+5-FU/LV; patients with an initial diagnosis of Ameri-can Joint Committee on Cancer, AJCC stage I-III are eligible if they developed metastasis after gemcitabine-based treatment

Reviewer 3:

Please also use a formula for albumin.

Author reply:

We are not sure of the meaning of “formula for albumin”. Would you please give more detailed information?

Reviewer 3:

Where the distant lymph node histopathological confirmed. If not leave them out of the table.

Author reply:

In most patients, the distant lymph node was not histopathologically confirmed. Therefore, we removed the distant lymph node in Table 1.

Reviewer 3:

Prior treatment group and operation history? This makes it very confusing for me. Since most patients had stage I-III disease it is surprising to see why just a few patients were operated on. Furthermore, the most performed surgical procedures were “Other procedures”.

Author reply:

Most patients (n=319) in our cohort had stage IV disease at diagnosis. Overall, 192 patients underwent surgery, including patients with stage IV disease who underwent palliative bypass surgery or only peritoneum biopsy because carcinomatosis was detected during laparotomy. Curative surgery was performed in 129 patients, including Whipple operation (n=77), distal pancreatectomy (n=42), and total pancreatectomy (n=10). Initially, we only listed the most common type of surgery and other curative surgery were classified into other procedures. We revised Table 1 to provide more detail about curative surgery.

Manuscript revision (Table 1):

Operation history

No surgery

74 (47.4%)

119 (64.0%)

88 (67.2%)

281 (59.4%)

Whipple operation

26 (16.7%)

34 (18.3%)

17 (13.0%)

77 (16.3%)

Distal pancreatectomy

20 (12.8%)

13 (7.0%)

9 (6.9%)

42 (8.9%)

Total pancreatectomy

3 (1.9%)

4 (2.2%)

3 (2.3%)

10 (2.1%)

Other procedures

33 (21.2%)

16 (8.6%)

14 (10.7%)

63 (13.3%)
